# Comparison of Five Models for Estimating the Water Retention Service of a Typical Alpine Wetland Region in the Qinghai–Tibetan Plateau

Meiling Sun [1,2,†], Jian Hu [1,2,†], Xueling Chen [1,2], Yihe Lü [3,4] and Lixue Yang [1,2,*]

1  Sichuan Zoige Alpine Wetland Ecosystem National Observation and Research Station, Southwest Minzu University, Chengdu 610041, China
2  Institute of Qinghai-Tibetan Plateau, Southwest Minzu University, Chengdu 610041, China
3  State Key Laboratory of Urban and Regional Ecology, Research Center for Eco-Environmental Sciences, Chinese Academy of Sciences, Beijing 100085, China
4  University of Chinese Academy of Sciences, Beijing 100049, China
*  Correspondence: yanglixue@swun.edu.cn; Tel.: +86-28-8552-3352
†  These authors have contributed equally to this work.

**Abstract:** Model evaluation of water retention (WR) services has been commonly applied for national or global scientific assessment and decision making. However, evaluation results from different models are significantly uncertain, especially on a small regional scale. We compared the spatial–temporal variations and driving factors of the WR service by five models (i.e., the InVEST model (InVEST), precipitation storage model (PRS), water balance model I (WAB I), water balance model II (WAB II), and NPP-based surrogate model (NBS) based on partial correlation analysis and spatial statistics on the Ramsar international alpine wetland region of the Qinghai–Tibetan Plateau (QTP). The results showed that the wetland area continued to decrease, and built-up land increased from 2000 to 2015. The average WR volume ranged from 2.50 to 13.65 billion $m^3 \cdot yr^{-1}$, with the order from high to low being the PRS, WAB I, WAB II, and InVEST models, and the average total WR capacity was $2.21 \times 10^9$ by the NBS model. The WR service followed an increasing trend from north to south by the InVEST, PRS, WAB I, and WAB II models, while the NBS model presented a river network pattern of high values. The WR values were mainly reduced from 2000 to 2010 and increased from 2010 to 2015 in the PRS, WAB I, WAB II, and InVEST models, but the NBS model showed the opposite trend. Precipitation determined the spatial distribution of WR service in the InVEST, PRS, WAB I, and WAB II models. Still, the spatial variation was affected by climate factors, while the NPP data influenced the NBS model. In addition, the InVEST model in estimating WR values in wetlands and the PRS and WAB I models poorly estimate runoff, while the WAB II model might be the most accurate. These findings help clarify the applicability of the WR models in an alpine wetland region and provide a valuable background for improving the effectiveness of model evaluation.

**Keywords:** alpine wetland; water retention service; models; Qinghai–Tibetan Plateau

## 1. Introduction

Ecosystem services (ESs) are the benefits people obtain from ecosystems which are essential for human well-being [1]. Water retention (WR) is a critical regulative service that refers to the water retained in ecosystems within a certain period [2–5]. Although China's total water resources rank sixth in the world, it has been experiencing severe water resource shortages due to climate change and rising water demands [6,7]. The degradation of WR service has accelerated water shortages and become one of China's major ecological problems [8]. China's first ecosystem assessment showed that WR service decreased from the southeast to the northwest inland areas in 2010 [3]. Then, a similar spatial pattern was also mapped from 2000 to 2013 in another way [9,10]. However, ESs are complex due

to strong scale effects, resulting in various spatial–temporal changes at small scales by models [11]. Therefore, it is necessary to compare the accuracy of WR service at a small regional scale with various models to better understand their applicability.

The Qinghai–Tibetan Plateau (QTP) is the world's largest water tower region, and poor availability and changes in its water services have been of great concern [12,13]. Researchers have mapped the spatial pattern of WR service by the Integrated Valuation of Ecosystem Services and Tradeoffs (InVEST) model, showing a decreasing trend from the southeast to the northwest of the QTP [14,15]. Several other WR models, initially developed in China, have also been applied to the QTP. The tradeoffs between carbon sequestration and WR service and the synergy between livestock production and WR service of alpine grassland were revealed by the water balance model I (WAB I) and the precipitation storage model (PRS), respectively [16,17]. The sensitivity of WR under future climate scenarios was quantified, and the ecological importance was mapped by the water balance model II (WAB II) [18,19]. In addition, the spatial–temporal characteristics of the water and nutrient retention service of the critical natural capital were displayed by the NPP-based surrogate model (NBS) [20]. However, few studies compared the results of models within different structures and characteristics of WR service, and it is necessary to reveal the WR model's performance for application in a specific region, especially for the spatial–temporal pattern and its changes, because the criteria of WR service by models are different, such as mm, $m^3$, or dimensionless. In addition, different results have been found in the Upper Upatoi Creek watershed and the Nansihu Lake basin between InVEST and the Soil and Water Assessment Tool (SWAT) [21,22]. Therefore, it is essential to compare various models about how well it fits applications to screen the model with strong applicability and low uncertainty to offer guidance on selecting more effective tools. The Zoige Plateau (ZP), in the eastern part of the QTP, is a critical WR region in the upper reaches of the Yellow River and is regarded as one of the most extensive alpine peatlands in the world for storing biotic carbon [23,24]. However, warming and drying climate trends and anthropogenic perturbations such as overgrazing and artificial ditch construction have resulted in the degradation of wetlands and a decrease in the runoff, threatening the stable supply of water resources [25–28]. Some researchers have evaluated the WR service of ZP [29], methane, and $CO_2$ emissions in the Zoige Wetland [30,31], and the ecosystem services value [32,33]. However, the accurate assessment of the spatial–temporal variation in WR service in the ZP remains a major challenge.

Researchers have evaluated the driving factors of WR service and highlighted that natural and human-induced factors are vital aspects. Additionally, these factors are independent and almost always multiple and interactive, so a one-to-one linkage between particular driving forces and particular ecosystem changes rarely exists [1]. In addition, topography, microclimate, vegetation, and hydrological processes also played vital role in soil water movement at a large scale [34]. Therefore, analyzing the combined effect of these drivers spatially to identify overlapping impacts on ESs is meaningful for spatial planning and management. Effective energy and mass transfer (EEMT), consisting of water, carbon, and energy, is a comprehensive climate indicator essential in controlling groundwater thickness and water availability [35,36]. It can be used for assessing the overlapping effects of climate-driving factors.

Therefore, our study considered ZP as the focus area and compared the WR service from 2000 to 2015 with five models, including the InVEST, PRS, WAB I, WAB II, and NBS. We used spatial statistical methods to reveal the changes in WR and the relationship with driving factors. The main objectives were to (1) quantify and compare spatial–temporal patterns and variations in WR service by five models under statistical methods, (2) reveal the relationship between natural and socioeconomic factors and WR service in different models by partial correlation analysis, and (3) discuss the applicability of five models to the alpine wetland region. This work will deepen our understanding of WR service simulation models and provide the theoretical basis for model application in an alpine wetland area of QTP.

## 2. Materials and Methods

### 2.1. Study Area

The ZP (31°50′–34°49′ N, 100°45′–103°39′ E) is located on the eastern margin of the QTP, with an area of 42,714 km² (Figure 1). It is a complete orbicular plateau surrounded by alpine mountains ranging from 2392 to 5059 m. Based on 39 years of data records (1980–2018), the average annual precipitation is approximately 712 mm, and the mean annual temperature is 0.87 °C. The main land types of ZP are grassland, shrubland, wetland, and forestland, accounting for 69.43%, 12.71%, 9.87%, and 5.23% of the total area in 2015, respectively. The proportion of cropland is relatively low, accounting for only 0.61% of the total area, and is mainly distributed in Aba County. The Zoige Wetland is an integral part of the Ramsar internationally important wetlands. In addition, almost all the rivers in this region belong to the Yellow River water system. Its tributaries mainly consist of the White River and Black River, providing at least 30% of the water flow into the upper Yellow River and becoming a vital water retention functional area [27]. It is crucial for China's ecological protection and high-quality development strategy in the Yellow River Basin.

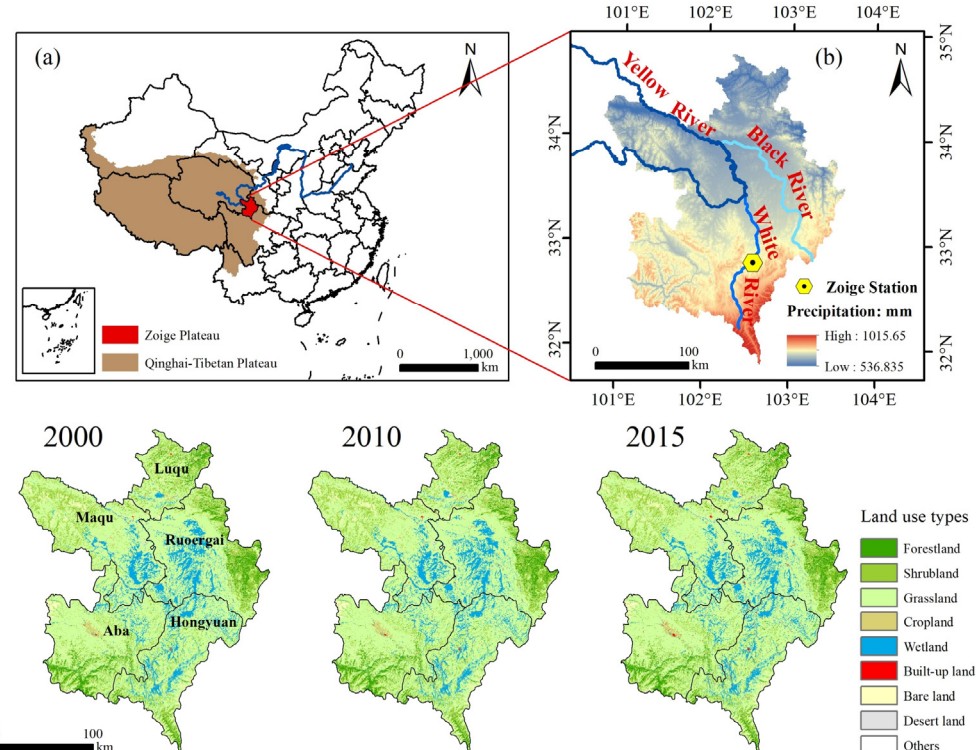

**Figure 1.** The location (**a**), mean annual precipitation from 1980 to 2018 (**b**), and land use type distribution from 2000 to 2015 of ZP.

### 2.2. Analysis Framework for Comparing the Water Retention Models

The purpose of this study was to compare the performance of WR service simulation models in an alpine wetland region, aiming to offer guidance on selecting and applying models in this area. Therefore, we selected five WR simulation models commonly used in QTP. After data preparation and preprocessing, we used the five models to map the spatial pattern of WR service from 2000 to 2015. Then, we used the spatial statistics method to compare the pattern of WR assessed by five models and its changes in the past 16 years. In addition, spatial partial correlation analysis was used to compare the relationship between WR assessed by different models and driving factors. Figure 2 shows the framework of our study.

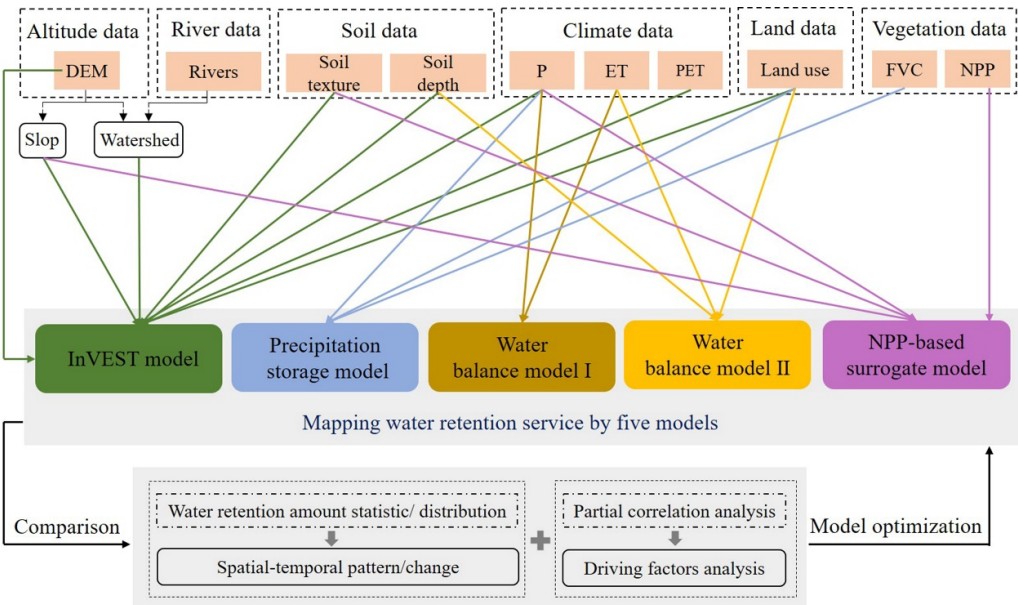

**Figure 2.** A framework for comparing water retention services by five models.

*2.3. Data Source and Processing*

The spatial data, including climate, soil, rivers, altitude, land use, vegetation, and socioeconomic data, are presented in Table 1. The land use and digital elevation model (DEM) data were obtained from the internal shared data of the Second Tibetan Plateau Scientific Expedition and Research. Origin 2022 software was used to draw the Sankey diagram of land use transfer. The gross domestic product (GDP) and population density (POP) data are raster data generated by interpolation based on GDP and population statistics data of all counties in China, considering land use types, night light brightness, residential density data, and spatial interaction with GDP. We spatially interpolated temperature and precipitation data from meteorological stations in China using Anusplin software at a 1 km resolution and then extracted them based on the study region. The resolution of all raster data was resampled to 30 m. In addition, we used the Penman–Monteith equation to calculate potential evapotranspiration (PET), and we obtained the annual PET by the sum of the monthly PET [37]. EEMT is the sum of energy input via effective precipitation and net primary production components; the calculation of EEMT ($J\,m^{-2}\,s^{-1}$ or $W\,m^{-2}$) is given in [35] as:

$$EEMT = E_{PPT} + E_{BIO} \tag{1}$$

$E_{PPT}$ is heat energy related to effective precipitation energy and mass transfer, and $E_{BIO}$ is NPP energy and mass transfer.

*2.4. Description of Selected Models*

2.4.1. InVEST Model

The InVEST model simulates and provides spatial information about ESs. The water yield module is based on the Budyko curve and water balance principles (Table 2). The following equation can calculate $WY(x)$:

$$WY(x) = \left(1 - \frac{AET(x)}{P(x)}\right) \times P(x) \tag{2}$$

where $WY(x)$ is the annual water yield for pixel $x$ (mm), $AET(x)$ is the potential annual evapotranspiration (mm), and $P(x)$ is the annual precipitation for pixel $x$ (mm). Taking average rainfall events from 1995–1999, 2005–2009, and 2010–2014 of four meteorological stations on the ZP as rainfall events in 2000, 2010, and 2015, the Z values were calculated as

follows: 11.59, 11.09, and 10.62 (Table S1). The *WR* values (mm) of each pixel were revised by the following formula [15]:

$$WR = min\left(1, \frac{249}{V}\right) \times min(1, 0.3TI) \times min\left(1, \frac{K_{sat}}{300}\right) \times WY \quad (3)$$

where *V* is the velocity coefficient; $K_{sat}$ is the saturated soil hydraulic conductivity (cm/d), calculated by Neuro Theta software according to the content of soil sand, silt, and clay (%); and *TI* is a topographic index, calculated from the following equation:

$$TI = \log\left(\frac{Drainage\_Area}{Soil\_Depth \times Percent\_Slop}\right) \quad (4)$$

where *Drainage_Area* is the number of catchment area grids, *Soil_Depth* is soil depth (mm), and *Percent_Slop* is the percentage slope.

### 2.4.2. Precipitation Storage Model (PRS Model)

The model refers to the decreased amount of water compared to bare land under the condition of rainfall generation as WR service [17]:

$$WR = A_f \times J_0 \times k \times \left(R_0 - R_f\right) \quad (5)$$

where $A_f$ is the area of ecosystem type (km²); $J_0$ is the annual precipitation (mm); *k* is the proportion of rainfall that can generate runoff, assigned a value of 0.6 in southern China; $R_0$ is the runoff ratio of bare land, assigned a value of 1; and $R_f$ is runoff ratio of the other land use types. The runoff ratio ($R_f$) of grassland was obtained from the vegetation cover ($f_c$):

$$R_f = -0.3187 f_c + 0.36403 \quad (6)$$

### 2.4.3. Water Balance Model I (WAB I Model)

The WR values (mm) are considered as the balance between precipitation (mm) and actual evapotranspiration (mm) [38]:

$$WR = P - ET \quad (7)$$

**Table 1.** Spatial data sources and description.

| Data | Spatial Resolution | Temporal Resolution | Units | Data Source |
|---|---|---|---|---|
| Rivers | 1:1,000,000 | 2019 | – | National Geomatics Center of China (http://www.ngcc.cn; accessed on 15 September 2021) |
| Soil texture/depth | 1 km | – | cm | China Soil Map-Based Harmonized World Soil Database (v1.1) (http://www.ncdc.ac.cn; accessed on 15 September 2021) |
| Temperature (T) | 1 km | 1980–2018, monthly | °C | |
| Precipitation (P) | 1 km | 1980–2018, monthly | mm | The National Meteorological Information Center of China (http://data.cma.cn/en; accessed on 20 September 2021) |
| Potential evapotranspiration (PET) | 1 km | 1980–2018, monthly | mm | |
| Evapotranspiration (ET) | 1 km | 2000–2018, 10 days | mm | [39] (https://www.sciencedirect.com/; accessed on 30 October 2021) |
| Net primary productivity (NPP) | 250 m | 2000–2015, monthly | 0.01 g/cm² | Institute of Remote Sensing and Digital Earth Chinese Academy of Sciences (http://eds.ceode.ac.cn/; accessed on 15 October 2021) |
| Fractional vegetation cover (FVC) | 250 m | 2000–2015, monthly | —— | |
| Gross Domestic Product (GDP) | 1 km | 2000–2015, yearly | Ten thousand yuan/km² | Resource and Environmental Science and Data Center (http://www.resdc.cn/;accessed on 15 October 2021) |
| Population density (POP) | 1 km | 2000–2015, yearly | People/km² | |

### 2.4.4. Water Balance Model II (WAB II Model)

This model refers to the *WR* (m$^3$) as the difference between the precipitation (mm) and the sum of actual evapotranspiration (mm) and surface runoff (mm), which is revised from the InVEST model [3].

$$WR = \sum_{i=1}^{n} A_i \times (P_i - R_i - ET_i) \times 10^3 \tag{8}$$

where *i* is the ecosystem, *n* is the number of ecosystem types, $A_i$ is an area of the ecosystem *i* (km$^2$), and $R_i$ is the surface runoff of the ecosystem *i*, obtained by multiplying precipitation and the surface runoff coefficient. $P_i$ and $ET_i$ are the precipitation and actual evapotranspiration of the ecosystem *i*.

### 2.4.5. NPP-Based Surrogate Model (NBS Model)

The NPP-based surrogate model simulate a variety of services, such as water retention, soil conservation, carbon sequestration, and biodiversity conservation [9,10,20]. The ability of WR (non-dimensional) service is calculated as follows:

$$WR = NPP_{mean} \times F_{sic} \times F_{pre} \times (1 - F_{slo}) \tag{9}$$

where $NPP_{mean}$ is the multi-year average net primary productivity, and $F_{sic}$, $F_{pre}$, and $F_{slo}$ are normalized (0–1) treated slope factor, soil infiltration capacity factor, and multi-annual average precipitation raster, respectively.

### 2.5. Analysis of Climate Change Trends

The linear regression method was adopted to analyze the variation trend of climate elements, including temperature, precipitation, evapotranspiration, and effective energy and mass transfer. The linear trend was detected by the least square regression method as follows:

$$yi = a + bxi + \varepsilon \tag{10}$$

where *a* is the intercept, *b* is the slope, and $\varepsilon$ is the residual. When *b* is negative, index *i* is a decreasing trend; when *b* is positive, index *i* is an increasing trend. MATLAB2019b and ArcGIS10.3 were used for trend analysis and spatial mapping, respectively. The change trend is significant when $p < 0.05$.

### 2.6. Analysis of WR Service Change and Drivers

### 2.6.1. Quantification of WR Values and Their Changes

After completing the WR service estimation by five models, we applied the random points tool to create 100 random points with no distance requirement to calculate the mean and standard deviation of the same model. At the same time, we removed the extreme values and retained 83 points in each model. Then, we carried out one-way ANOVA and multiple comparisons based on the LSD test to investigate the differences in model results. We used the raster calculator tool to calculate the spatial change in WR by subtracting layers of the same model from two years to obtain the number of change values on each pixel.

### 2.6.2. Determination of Land Use Attributes and Changes

Humans influence ecosystem structure and function by changing land characteristics and enhancing utilization to meet their demand for land supply capacity. In this study, we used land use intensity to characterize the spatial–temporal changes in land use attributes [40]:

$$L = \sum_{i=1}^{n} A_i \times C_i = \sum_{i=1}^{n} A_i \times \frac{S_i}{S} \tag{11}$$

where $L$ is land use intensity index; $A_i$ is the grade index of land use type $i$. Considering the intense grazing activities in the study area, grassland is set to 2.5, the forestland, shrubland, and wetland are all 2, cropland is 3, built-up land is 4, and other types is 1. $C_i$ is the area percentage of land type $i$, $S_i$ is the area of the land use type $i$, and $S$ is the total land area of the study area.

**Table 2.** Basic characteristics of the five selected models.

| Model | Types | Spatial Scale | Temporal Scale | Advantages and Disadvantages | Reference |
|---|---|---|---|---|---|
| InVEST model | Water balance-based | Watershed | Year | (1) Strong visualization and dynamic (2) Complicated data input (3) Ignores interaction between surface water and groundwater | [14,15,21] |
| PRS model | Process-based | All scales | All scales | (1) Uncertainty of runoff and rainfall parameters (2) Relative value to bare ground, not an absolute value | [17,26,41] |
| WAB I model | Water balance-based | All scales | All scales | (1) Easy operation (2) Ignores runoff and groundwater (3) Suitable for dry regions | [38] |
| WAB II model | Water balance-based | All scales | All scales | (1) Easy operation (2) Uncertainty of runoff parameters | [3,19] |
| NBS model | Surrogate biophysical indicators-based | Regional scale | Year | (1) Not applicable to water bodies; (2) Affected greatly by NPP data (3) Cannot give specific physical quantities | [9,10] |

### 2.6.3. Exploration of Factors Influencing WR Service

We selected precipitation (P), evapotranspiration (ET), temperature (T), and EEMT as the natural driving factors and GDP, POP, and land use intensity (L) as the socioeconomic driving factors. Then, we used partial correlation analysis to explain the drivers related to WR service. The data were normalized first using the Z score method, and the Pearson correlation was employed to calculate the simple relationship between a dependent variable and a single independent variable. Then, the partial correlation analysis was applied when the two variables were simultaneously related to other variables (Equation (12)).

$$R_{xy,z} = \frac{r_{xy} - r_{xz}r_{yz}}{\sqrt{(1 - r_{xz}^2) \times (1 - r_{yz}^2)}} \tag{12}$$

where $R_{xy,z}$ is the partial correlation coefficient between $x$ and $y$, excluding the impact of $z$; $r_{xy}$ is the correlation coefficient between $x$ and $y$; $r_{xz}$ is the correlation coefficient between $x$ and $z$; and $r_{yz}$ is the correlation coefficient between $y$ and $z$.

A $T$-test was used for the reliability of the results of the partial correlation analysis:

$$t = \frac{r_{xy,z}}{\sqrt{1 - r_{xy,z}^2}} \sqrt{n - m - 1} \tag{13}$$

where $n$ is the number of years and $m$ is the number of the independent variables. The significance level was at 0.05.

## 3. Results

### 3.1. Land Use Change from 2000 to 2015

Grassland, shrubland, forestland, wetland, and cropland were the predominant land use types, accounting for 98% of the study region (Figure 3). From 2000 to 2010, grassland increased by 109.78 km$^2$, while wetland and cropland decreased by 112.01 and 8.64 km$^2$, respectively. The shrubland and forestland increased by 3.10 and 0.65 km$^2$, respectively. From 2010 to 2015, grassland, shrubland, forestland, and cropland decreased by 72.49, 17.55, 3.03, 0.65, and 9.15 km$^2$, respectively, while built-up land, bare land, and desert land increased by 81.86, 14.62, and 6.41 km$^2$, respectively. Over the past 16 years, the wetland area decreased the most, reaching 129.56 km$^2$, and was mainly converted to grassland, bare land, and built-up land. Except for a slight increase (3.98 km$^2$) in Luqu County, all the other counties decreased, and Ruoergi County decreased the most, by 75.37 km$^2$ (Figure S1). In contrast, built-up land increased for all counties, reaching 105.26 km$^2$, mainly from grassland, cropland, and wetland. Grassland increased by 37.28 km$^2$ and was distributed primarily in Ruoergai and Maqu counties. Desert land increased by 10.11 km$^2$ mainly from grassland degradation. In addition, shrubland, forestland, and other land use types remained stable.

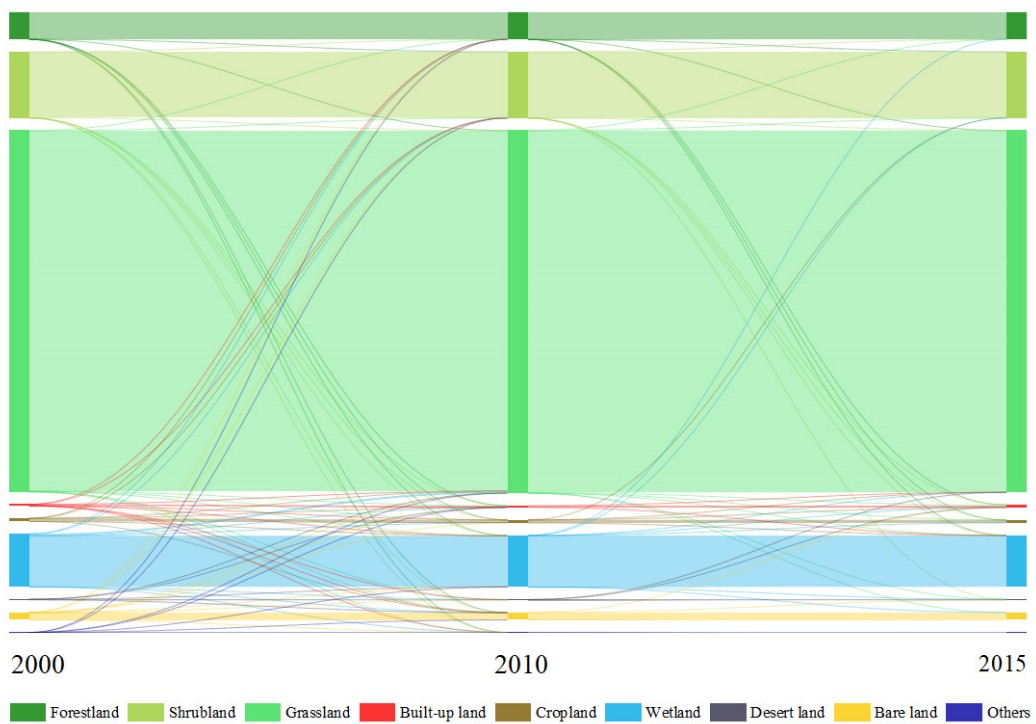

**Figure 3.** Land use composition and transformation from 2000 to 2015.

### 3.2. Climate Change Trends

From 1980 to 2018, the temperature showed a significant increasing trend ($p < 0.05$), with Luqu, Ruoergi, and Maqu counties in the north being lower than Aba and Hongyuan counties in the south and the highest increase in northwestern Maqu County (Figure 4). Precipitation decreased at approximately 42% of the regions ($p > 0.05$), mainly in northwestern Aba County, southeastern Hongyuan County, central and eastern Ruoergi County, and southern Luqu County. From 1980 to 2018, evapotranspiration showed a significant decreasing trend outside the southeast, northern, and eastern of ZP ($p < 0.05$), showing a northwest–southeast spatial distribution pattern. In addition, EEMT showed a significant increasing trend ($p < 0.05$) in other areas outside the southwest of Maqu County, parts of Hongyuan County, and Aba County.

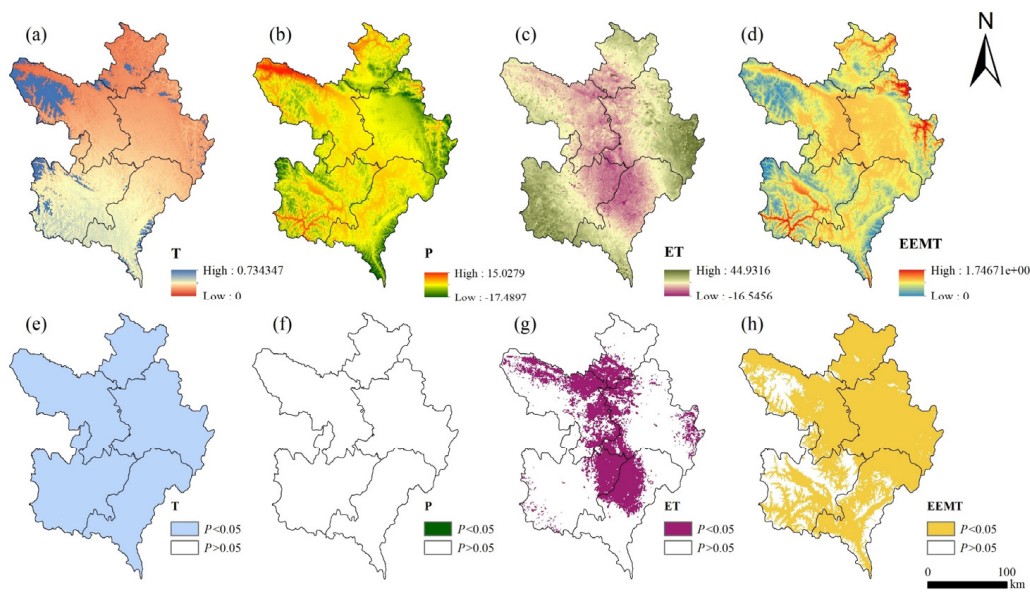

**Figure 4.** Spatial distribution of change trend values (**a–d**) and significance (*p* < 0.05) (**e–h**) of temperature (T), precipitation (P), evapotranspiration (ET), and effective energy and mass transfer (EEMT) of ZP.

### 3.3. Spatial Pattern of WR Service

The spatial pattern of the WR service showed substantial heterogeneity across the ZP. The spatial similarity was present between different models with an increasing trend from north to south in the InVEST, PRS, WAB I, and WAB II models over the past 16 years. However, the high values spread in a river network spatial pattern in the NBS model were consistent with the distribution of river channels (Figure 5). From the pixel value perspective, the maximum value in the WAB I model was 901.24 mm in 2000. The InVEST, WAB I, and WAB II models always had the largest weight of <100 mm, averaging approximately 83.61%, 43.03%, and 71.87%, respectively. The total area of pixel values ≤300 mm was approximately 93.35–99.58% of the ZP during the study period for these three models (Table S2). The PRS model had a maximum pixel area of 300–400 mm, with an area ratio of 56.04–57.21%. More than 96% of the region's WR ability by the NBS model was below 100 in the research period. In addition, all models showed that grassland was the land type with the highest WR capacity, followed by shrubland, forestland, wetland, and other land use types (Table S3).

### 3.4. Temporal Change in WR Service

The InVEST model had the lowest values of total WR volume, with a mean value of only 2.50 billion m$^3$·yr$^{-1}$ (Figure 6a). The PRS model always provided the highest WR volume, approximately 13.65 billion m$^3$·yr$^{-1}$. The results of the WAB I and WAB II models behaved between the above two models but were closer to the InVEST model, about 5.71 and 3.16 billion m$^3$·yr$^{-1}$, respectively. In addition, except for the NBS model, the WR values per grid of the PRS model were significantly higher than those of the other three models. In contrast, the InVEST model was markedly lower than PRS, WAB I, and WAB II models (Figure 6b) (*p* < 0.05). From 2000 to 2010, WR in the InVEST, WAB I, and WAB II models had a decreasing trend. The InVEST model reduced the least by 0.56 billion m$^3$; no decreasing regions were concentrated in the northwestern Maqu and Luqu counties, at 4493.52 km$^2$ (Figure 7 and Table S4). The WAB I model reduced the most, reaching 1.08 billion m$^3$. The unreduced area was only 2795.66 km$^2$, distributed in the junctional area of Ruoergai, Maqu, and Luqu counties. The WAB II model reduced 1.05 billion m$^3$, and the spatial variation pattern was similar to the WAB I model. However, the area unreduced was more extensive at 3111.24 km$^2$. The PRS model showed a slight increase of 0.11 billion m$^3$, and there was also an evident increase in wetland area compared to the

InVEST model, reaching 9231.54 km$^2$ overall. From 2010 to 2015, the above four models showed an increasing trend. Maximum and minimum increased values appeared in the PRS and InVEST models at 1.14 and 0.09 billion m$^3$, respectively. In addition, the WAB I and WAB II models showed 0.78 and 0.67 billion m$^3$ increases, respectively. It was mainly in the southwestern margin of Aba County and the eastern marginal regions of Ruoergai County for reduced areas. From 2000 to 2015, similar spatial–temporal patterns appeared between the InVEST and PRS models and the WAB I and WAB II models. The InVEST model experienced an increase of 0.03 billion m$^3$, with an area of 8203.94 km$^2$ detected to decrease in southwestern and southeastern margin areas, including Ruoergai, Hongyuan, and Aba counties. In comparison, the PRS model increased by 1.25 billion m$^3$ with a larger reductive area at 14,580.12 km$^2$, including most of Hongyuan and Aba counties and the southeastern part of Ruoergai County. The WAB I and WAB II models estimated reductions of 0.30 and 0.38 billion m$^3$, covering 24,613.58 and 24,970.92 km$^2$, respectively. A northwest–southeast distribution was found for unreduced regions. The InVEST, PRS, WAB I, and WAB II models all had more than 96.40% of the pixels whose variation in WR ranged within 200 mm from 2000 to 2015. For the NBS model, the average WR capacity per pixel was 46.93, and the total was $2.21 \times 10^9$. In addition, the spatial variation in the NBS model behaved differently from the other four models. There was an increasing trend from 2000 to 2010 and a decreasing trend in the central and easternmost parts. The majority decreased from 2010 to 2015 and were distributed in the region outside Luqu County for unreduced values. Over the past 16 years, the reduced pixels spread in much of the central and eastern regions, at 42,414.28 km$^2$. The total WR capacity changed slightly to 0.055 billion.

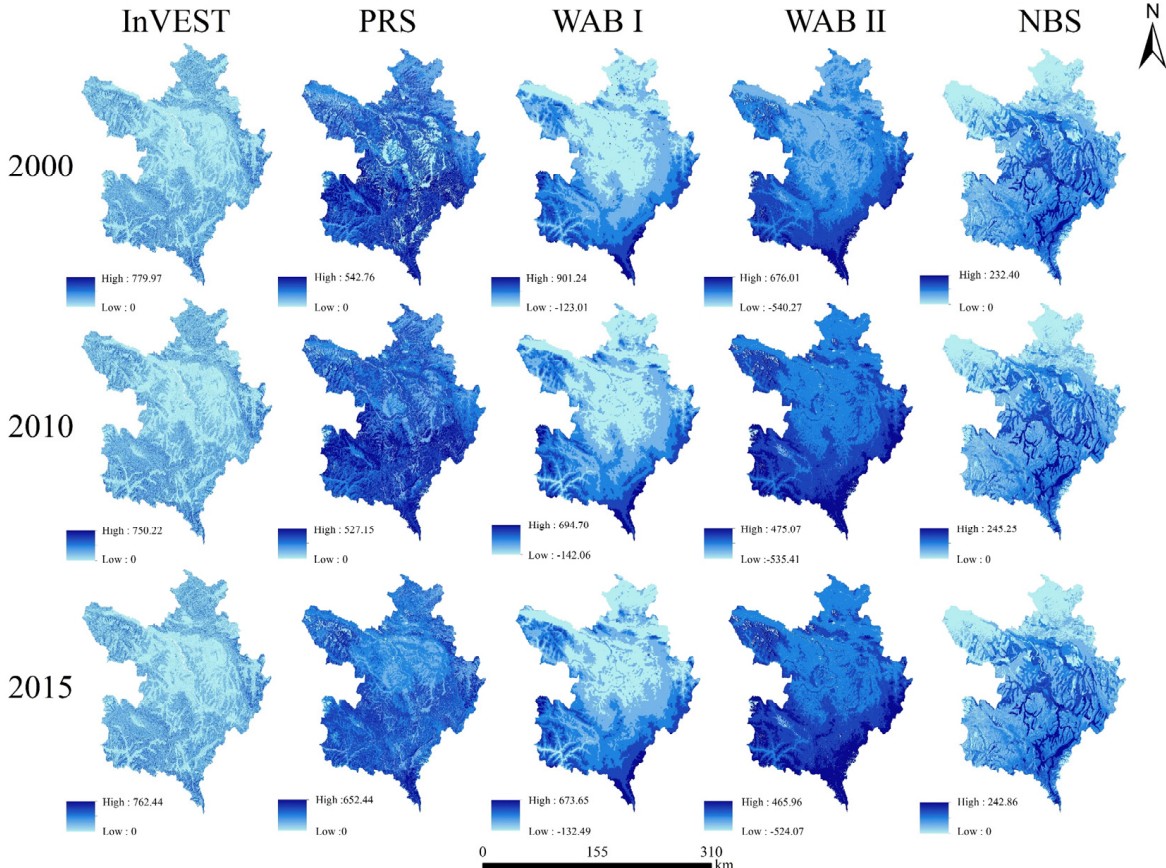

**Figure 5.** Spatial distribution of WR service by models from 2000 to 2015. The units for the InVEST, PRS, WAB I, and WAB II models are mm/grid and dimensionless for the NBS model.

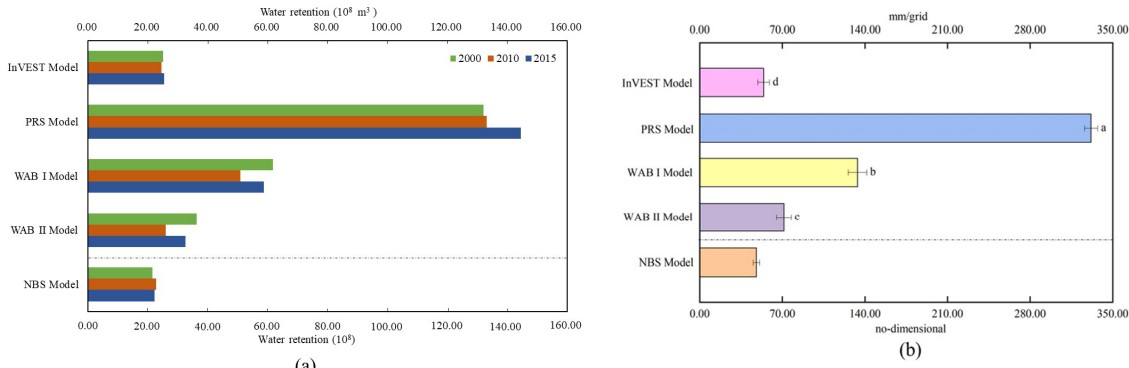

**Figure 6.** Characteristics of water retention volume or capability by five methods from 2000 to 2015. The units for the InVEST, PRS, WAB I, and WAB II models are m$^3$, and dimensionless for the NBS model (**a**). The values are the mean $\pm$ standard error, and lowercase letters indicates significant differences among the different models ($p < 0.05$) (**b**).

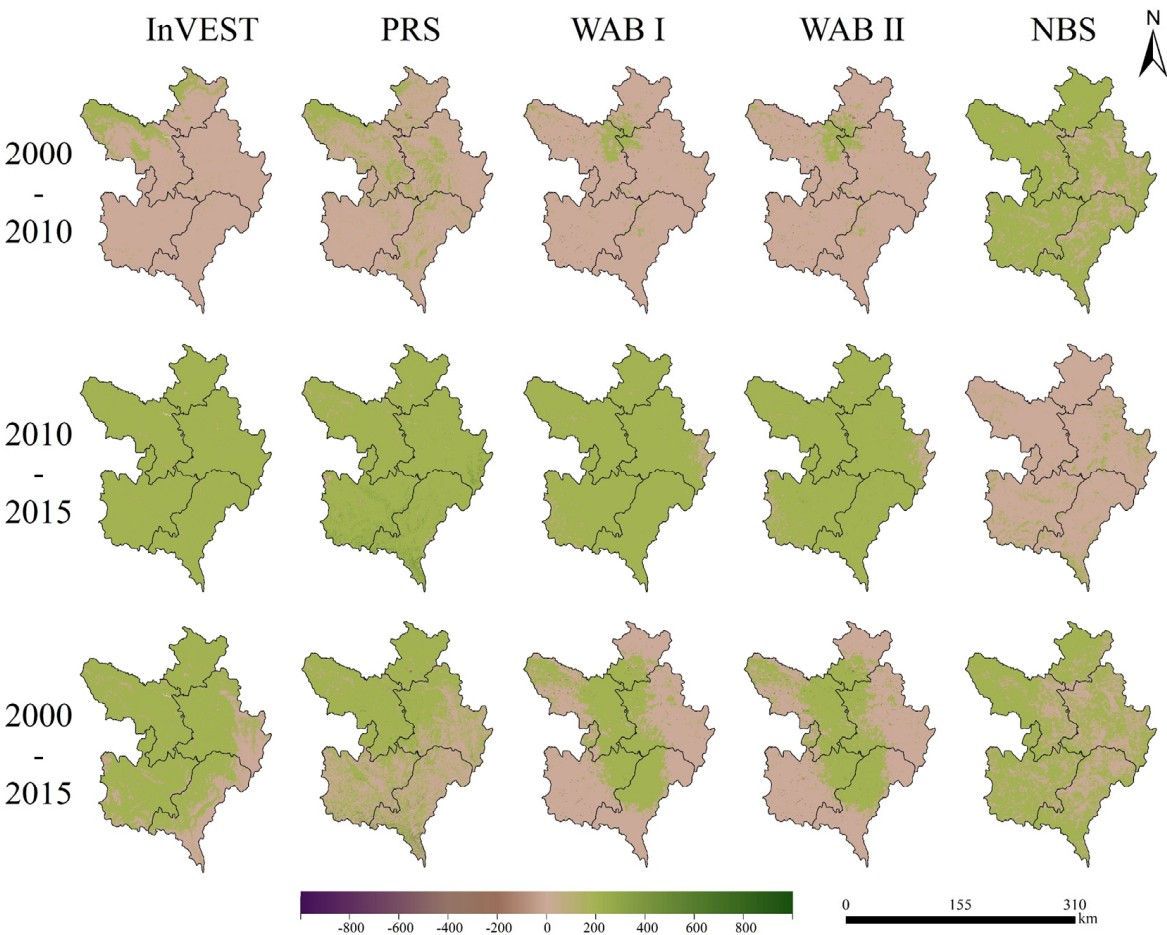

**Figure 7.** The spatial–temporal variations in WR service by five models in different periods. The units for models except the NBS model are mm/grid, and dimensionless for the NBS model.

### 3.5. The Driving Factors for WR Service Change

#### 3.5.1. Influence of Climatic Factors on the WR Service

The WR services simulated by the five models correlated significantly with climatic factors and geographic characteristics (Figure 8 and Table 3). The results of the InVEST model showed that 56.88% of the study area had a positive relationship between WR and temperature, concentrated in Aba, Hongyuan, and Luqu counties. At the same time, the negative correlation regions were distributed in most of Maqu, Ruoergai, and the low

areas in the southern mountains. In addition, the largest area with a positive relationship between WR and precipitation was found in the InVEST model, accounting for 65.90%, distributed in the vast area outside the southernmost parts of Aba and Hongyuan counties. In addition, WR was positive with evapotranspiration, mainly in central Hongyuan County, the northern margin of Luqu County, northwestern Maqu County, and northeastern Aba County. There was a significant negative correlation between WR and EEMT in the northern part of Luqu County, the eastern margin of Ruoergai County, and most of Hongyuan County by the InVEST model. The WAB I and WAB II models assumed an almost consistent spatial characteristic between WR service and four climatic factors. In detail, WR was positively correlated with temperature in the junctional region of Luqu, Maqu, and Ruoergai counties and southeastern Aba County, at approximately 41.89% and 49.65% of the total area, respectively. Likewise, the WR of these two models had a positive relationship with precipitation in northwestern Maqu County. However, the WR of the WAB I model in the western part of Aba County and the discontinuous central part of ZP showed a positive relationship with precipitation. This relationship existed in the WAB II model on the eastern margin of Ruoergai County. In addition, WR was almost negative with evapotranspiration and positive with EEMT for the WAB I and WAB II models.

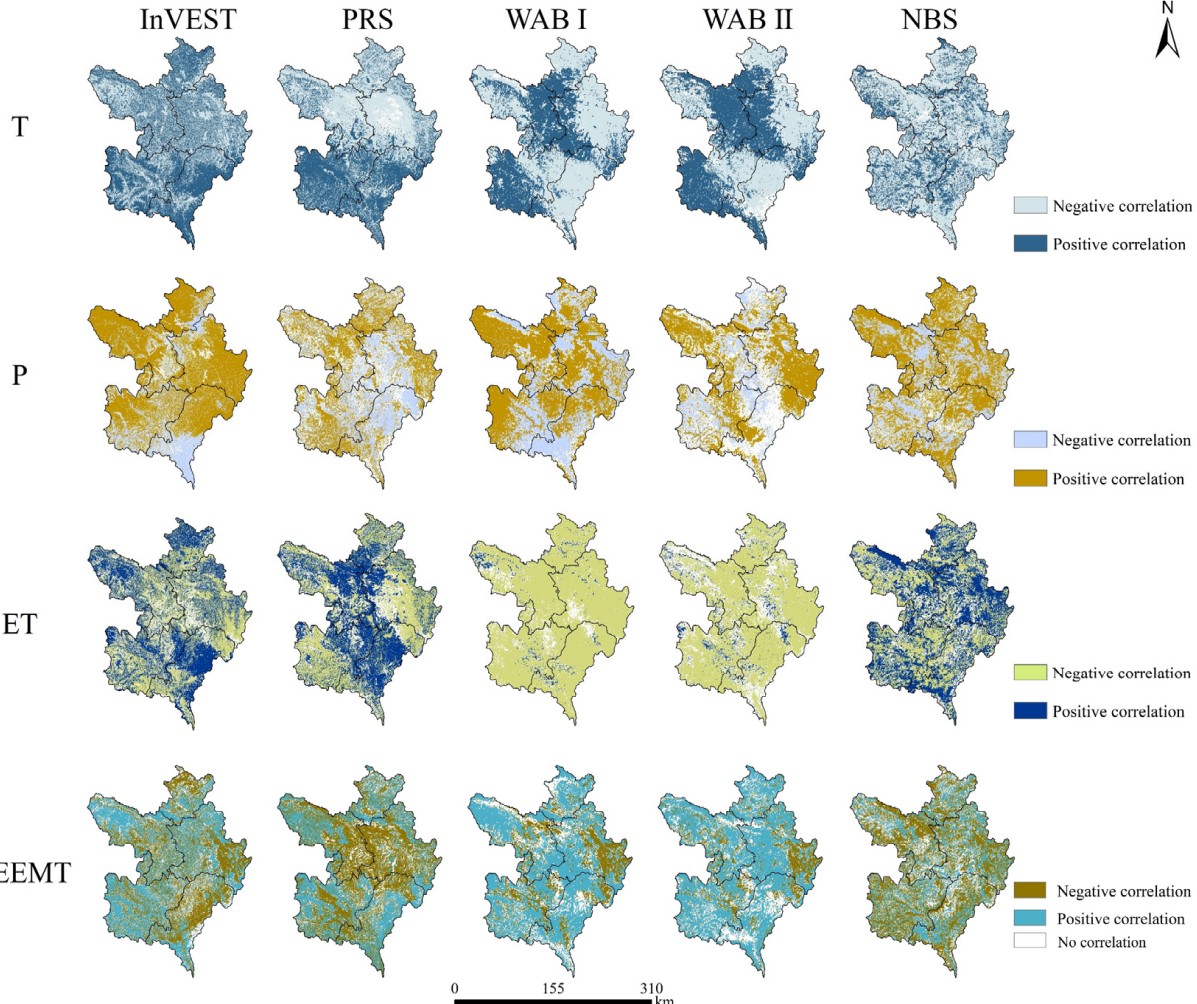

**Figure 8.** Distribution of partial correlation between natural drivers and WR service by five models ($p < 0.05$). T: temperature, P: precipitation, ET: evapotranspiration, EEMT: effective energy and mass transfer.

**Table 3.** Area and percentage of positive and negative correlations between water retention and drivers.

| Drivers | | Temperature | | Precipitation | | Evapotranspiration | | EEMT | | Land Use Intensity | | GDP | | Population Density | |
|---|---|---|---|---|---|---|---|---|---|---|---|---|---|---|---|
| Models | | Area (km²) | Percent (%) | Area (km²) | Percent (%) | Area (km²) | Percent (%) | Area (km²) | Percent (%) | Area (km²) | Percent (%) | Area (km²) | Percent (%) | Area (km²) | Percent (%) |
| InVEST model | Positive | 24,294.96 | 56.88 | 28,149.75 | 65.90 | 17,523.39 | 41.03 | 18,773.75 | 43.95 | 11,741.67 | 27.49 | 15,585.16 | 36.49 | 16,831.87 | 39.41 |
| | Negative | 14,935.75 | 34.97 | 9061.39 | 21.21 | 17,229.58 | 40.34 | 15,292.87 | 35.80 | 10,721.12 | 25.10 | 13,114.63 | 30.70 | 12,258.14 | 28.70 |
| PRS model | Positive | 18,583.70 | 47.65 | 16,580.52 | 38.82 | 18,159.81 | 42.52 | 19,017.32 | 44.52 | 10,712.33 | 25.08 | 11,966.04 | 28.01 | 15,317.53 | 35.86 |
| | Negative | 20,415.02 | 52.35 | 15,123.32 | 35.41 | 16,899.88 | 39.57 | 16,638.75 | 38.95 | 3728.08 | 8.73 | 16,747.22 | 39.21 | 12,381.78 | 28.99 |
| WAB I model | Positive | 17,892.73 | 41.89 | 23,248.45 | 54.43 | 1155.51 | 2.71 | 24,399.69 | 57.12 | 9630.13 | 22.55 | 14,385.66 | 33.68 | 15,850.65 | 37.11 |
| | Negative | 21,615.77 | 50.61 | 15,096.95 | 35.34 | 36,965.40 | 86.54 | 8498.39 | 19.90 | 11,715.66 | 27.43 | 14,524.02 | 34.00 | 15,063.18 | 35.27 |
| WAB II model | Positive | 21,209.20 | 49.65 | 18,129.24 | 42.44 | 2259.73 | 5.29 | 26,194.72 | 61.33 | 10,960.71 | 25.66 | 13,685.20 | 32.04 | 16,223.72 | 37.98 |
| | Negative | 16,417.89 | 38.44 | 9235.47 | 21.62 | 30,617.34 | 71.68 | 6485.74 | 15.18 | 13,304.71 | 31.15 | 15,789.56 | 36.97 | 15,458.57 | 36.19 |
| NBS model | Positive | 14,348.02 | 33.59 | 23,972.37 | 56.12 | 18,703.74 | 43.79 | 13,826.34 | 32.37 | 13,210.07 | 30.93 | 17,309.52 | 40.52 | 15,799.46 | 36.99 |
| | Negative | 19,906.80 | 46.61 | 12,246.80 | 28.67 | 16,416.53 | 38.43 | 18,614.40 | 43.58 | 11,656.09 | 27.29 | 12,045.92 | 28.20 | 15,446.70 | 36.16 |

Similar to the InVEST model, WR service was positively correlated with temperature in Aba and Hongyuan counties for the PRS model. In contrast, the other three counties were opposite. There was an apparent negative relationship between WR and precipitation in the central areas of ZP and most of Hongyuan County. Evapotranspiration positively impacts the WR in a large area at the junction of Maqu, Luqu, and Ruoergai counties, northwestern Aba County, and most of Hongyuan County. In addition, the WR of the PRS model service in the central parts of ZP had a significant negative correlation with EEMT than the other four models. The NBS model was spatially different from the other four models. There was an apparent negative correlation between WR capacity and temperature. The WR capacity positively correlated with precipitation outside the northeast of Maqu County, most of Aba County, and central Hongyuan County. The NBS model also differed from the other four models between WR service and evapotranspiration, with a negative correlation mainly in southwestern Maqu County, northwestern Aba County, and Hongyuan County. The T-test performed the same as the InVEST, WAB I, and WAB II models in the eastern margin of Ruoergai County and differently from the PRS model in the central parts of ZP for the negative performance between WR service and EEMT.

### 3.5.2. Influence of Socioeconomic Factors on WR Service

The WR of the InVEST model showed a positive relationship with land use intensity in Hongyuan County and the eastern margin of Ruoergai County (Figure 9 and Table 3). In addition, the InVEST model showed that GDP significantly inhibited WR service in 39.49% of areas, including southeastern Maqu County and Aba and Hongyuan counties. However, the WR of the five models positively correlated with population density. The WR of the InVEST model indicated a significant negative correlation with population density at a ratio of 28.70% in southwestern Aba County and northeastern Luqu and Ruoergai counties. The WR services by the WAB I and WAB II models all showed a negative correlation with land use intensity in the central parts of ZP and a positive correlation in southern and northern Hongyuan County. Unlike the InVEST model, both models showed that WR services were positively correlated with Hongyuan County's GDP. At the same time, they were negatively correlated with GDP in the eastern parts of Ruoergai County and most of Luqu County. In addition, population density also significantly inhibited WR service in the northeastern part of Aba County and the central part of Hongyuan County.

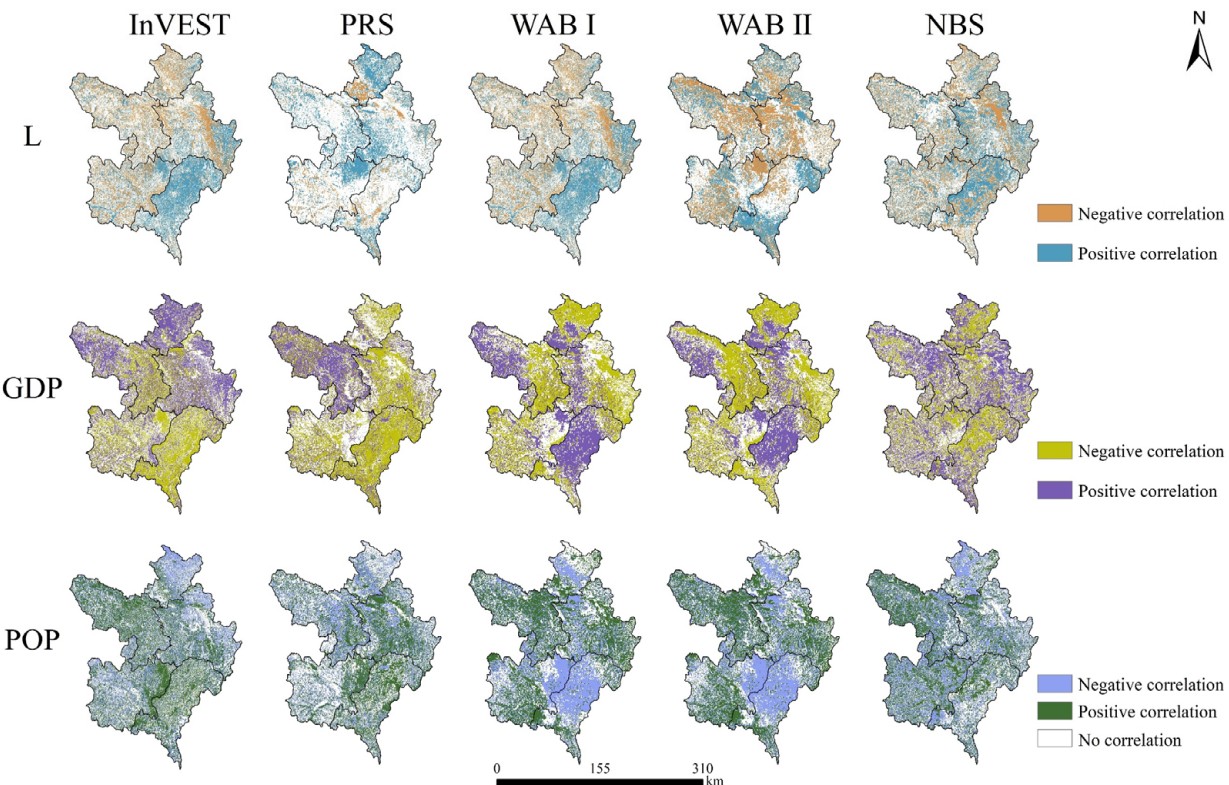

**Figure 9.** Distribution of partial correlation between socioeconomic factors and WR service by five models (*p* < 0.05). L: land use intensity, GDP: gross domestic product, POP: population density.

The WR of the PRS model positively correlated with land use intensity in the central part of ZP and northern Luqu County. The GDP in Maqu County and population density in the central part of ZP increased the WR of the PRS model. In addition, the WR capacity of the NBS model was equally as positive with land use intensity in Hongyuan and Ruoergai counties as of the InVEST model. In addition, WR capacity had the least negatively correlated area with the GDP, mainly in the northeastern part of Hongyuan County and the southeastern part of Luqu County. Furthermore, most central and eastern areas represented a negative relationship between WR ability and population density.

## 4. Discussion

### 4.1. Model Uncertainty and Applicability

The WAB I model takes the difference between precipitation and evapotranspiration as the WR service, which is consistent with the principle of the water yield module in the InVEST model. However, 30% of precipitation flows out as runoff in the growing season, so neither the WAB I model nor the water yield volume can be used as the actual WR volume of ZP [25,27]. The WAB I model and water yield module are more suitable for arid and semiarid areas where water budgets mainly depend on precipitation and evapotranspiration [38]. In addition, WR is the infiltration amount of the soil layer after precipitation minus evapotranspiration and surface runoff [42]. Therefore, the InVEST and WAB II models take soil infiltration over time as the WR service. However, the total WR volume of wetlands was the lowest by the InVEST model. The main reason was that many pixel values of lakes and rivers were zero by the water yield module and the InVEST model [15]. Therefore, the InVEST model may have uncertainty in assessing WR service volume in this study. In addition, the WR of the PRS model was mainly affected by precipitation and the parameters input to the model. We used the general parameter "0.6" as the proportion of precipitation that can generate runoff. However, observation data showed that evapotranspiration accounts for more than 60% of the annual precipitation,

meaning that the amount of water stored in an ecosystem is less than or equal to 40% [27,41]. Therefore, the actual volume of water storage should be lower than estimated. The WAB II model may be the most accurate for WR volume despite five WR models showing a consistent spatial pattern. Furthermore, the primary indicator of the NBS model is NPP data, contributing the most to the WR values [10]. Herbaceous marsh accounted for 91.43–92.20% of wetland areas from 2000 to 2015. The high value of NPP was more extensive along rivers and wetland vegetation growth areas, where herbaceous peat bog soils are abundant [43]. However, the ZP has higher vegetation cover, prosperous plant communities, and ecosystem types than other regions, as well as a complex plateau landform landscape. Water bodies with vegetation cover were easily classified as vegetation in remote sensing data, which could cause uncertainty in the results of the NBS model [44,45]. Vegetated marshlands are probably the only spectrally unique wetland category that can be discerned from TM and ETM+ imagery [46]. Furthermore, monitoring suffered due to frequent variable weather and thick clouds, resulting in a lack of long–time series data in the QTP [45]. Therefore, high temporal and spatial resolution remote sensing products on alpine ecosystems, including land use and vegetation characteristics data, contribute to the WR service simulation.

### 4.2. Spatial–Temporal Patterns of WR Service

We investigated the WR service's spatial–temporal patterns from 2000 to 2015 with five WR models. The annual WR service had similar spatial patterns to the distribution characteristics of precipitation in the InVEST, PRS, WAB I, and WAB II models. The mountainous southern areas receive more rain than the plains, and the distribution of precipitation across the region increases gradually from >500 mm in the north to >1000 mm in the south. In addition, these four models also displayed higher values distributed in the high-altitude regions, with the central and northern regions having lower values, consistent with previous studies [29]. Under limited spatial data conditions, the WAB I model best shows the WR's spatial pattern since only precipitation and evapotranspiration raster data are required as inputs. From 2000 to 2015, the increased region by the WAB I and WAB II models performed a northwest–southeast distribution. Differing from the study that increased in the Aba and Ruoergai counties from 2000 to 2017, and this may be due to the different data sources and change analysis methods [29]. In addition, the WR of the InVEST and PRS models increased widely in the northwest of ZP. The results showed that the WR's changing area (increase or decrease) calculated by the InVEST model was more significant than that of the WAB II model [15,18]. High WR values are presented in river network patterns by the NBS model, mainly affected by the distribution of the NPP value of herbaceous wetlands [43]. From 2000 to 2015, wetlands' total WR ability decreased because the entire marsh NPP dropped [47]. Therefore, this model is suitable for identifying the WR capacity of herbaceous wetlands.

### 4.3. Driving Forces of WR Service

WR service is one of the most critical regulation services in ecosystems. It can be affected by climate change and anthropogenic perturbation. A warming and drying trend has occurred in the upper Yellow River, and increased evapotranspiration has caused water loss due to rising temperature [41]. The primary relationship between WR service and precipitation was positive for all five models and determined the spatial–temporal pattern in the InVEST, PRS, WAB I, and WAB II models. Precipitation decreased in the eastern and southern parts of the ZP, consistent with the WR service's spatial changes by InVEST from 2000 to 2015. Therefore, precipitation remains this model's critical driver of WR service changes [15]. In addition, the WAB I and WAB II models showed negative correlations with evapotranspiration over 71% area of ZP. The variation in WR values was consistent with the spatial variation in evapotranspiration. Therefore, the spatial variation in the WR service in these two models was mainly related to evapotranspiration. WR of the PRS model had a positive relationship with evapotranspiration. The evapotranspiration decreases, and

the water storage of the ecosystem increases. Peatlands with carbon-rich soils have been characterized in convergent areas of greater EEMT and positively correlated with annual baseflow contributing to headwater stream runoff [48–50]. EEMT was negative in most wetland parts, mainly in the western Ruoergai County and the eastern Maqu County, by the PRS, InVEST, and NBS models, while positive in the WAB I and WAB II models. The WAB I and WAB II models may better reveal the wetland region's climate drivers. The reason may be that the PRS model evaluates the capacity of an ecosystem to regulate water, while the WAB I and WAB II models simulate the water stock of the ecosystem, both above and below ground [51]. Baseflow expressed an increasing trend with climate warming, which means water storage may become smaller [50]. Furthermore, the InVEST model needs to estimate the wetland accurately [15]. The correlation distribution between the NBS model and climate factors existed consistently, indicating that WR services were affected mainly by NPP data input and its changes [9,47].

Warming and drying trends may convert wetlands to grassland, but overgrazing, gully drainage, and peat mining are the main factors [52,53]. Since the 1960s, nearly 1000 drainage channels have been built in the Zoige wetlands, which drain groundwater, slow subsurface flow except in the rainy season, reduce the quality and area of wetlands, and lower the adjacent water table [25,47,49]. Researchers have shown that the mean annual runoff depth from the White (Black) River decreased by 28% (35%) in the human-induced period [25]. From 2000 to 2015, 129.56 $km^2$ of wetland areas mainly transformed into grassland, bare land, and built-up land, which altered water purification processes and weakened the capacity of soil to regulate and store water. Furthermore, this region implemented many measures and policies, including rotational and restricted grazing, filling ditches, planting grasses, and forbidding peat mining [47,54]. Afterward, the amount of wetland, grassland, and forestland areas increased, which can help to conserve water and soil [55]. Compared to natural factors, the relationship between socioeconomic factors and WR services is relatively weak, possibly due to the small spatial scale of the study area [11]. However, these human disturbances should be considered in future WR service evaluations.

In addition, precipitation, evapotranspiration, infiltration, and runoff impact the water budgets in watersheds [56]. The White River and Black River watersheds, the basic unit of the Earth's land surface system, constitute important WR functional areas in the upper Yellow River [57]. Additionally, the region is transitioning between seasonally frozen ground and permafrost. The boundaries between seasonally frozen land and permafrost are changing due to temperature increases, which impact the downstream water supply [41]. For a cold region, glaciers, snowmelt, and permafrost (frozen water components) are also essential parts of the water budget. However, the models discussed in this study may not have an adequate process to capture those budgets. Beyond that, various uncertainties exist in remote sensing data, especially in alpine regions; therefore, an integrated, distributed, and multiscale observation dataset is essential [44,45]. Comprehensive management of the Heihe River Basin in China through coupling ecohydrological and socioeconomic models undoubtedly provided scientific methods and ideas for watershed management in the Zoige Plateau.

## 5. Conclusions

This study compared five WR service evaluation models in terms of spatial–temporal patterns and driving factors from 2000 to 2015 based on spatial analysis and statistical methods. The InVEST, PRS, WAB I, and WAB II models showed similar spatial patterns of an increasing trend from north to south determined by the spatial distribution of precipitation. The WAB II model might be the most suitable because the InVEST model underestimates WR service due to weakening simulation in the wetland regions. The PRS model overestimates the proportion of precipitation that can generate runoff. The WAB I model lacks runoff procession but is the most convenient for showing a spatial pattern. The WAB I and WAB II models were better at revealing the overlapping effects of climate drivers in the wetland regions. However, the NBS model presented a river network pattern of high values

over the past 16 years. More accurate NPP spatial data in identifying herbaceous marsh areas are essential for the assessment results of the NBS model. In addition, artificial ditch construction and ecological restoration measures should be included in evaluating WR services. The development of integrated watershed models in the alpine wetland region is needed.

**Supplementary Materials:** The following supporting information can be downloaded at: https://www.mdpi.com/article/10.3390/rs14246306/s1, Figure S1: Land use composition and transformation of Aba County (a), Hongyuan County (b), Luqu County (c), Maqu County (d) and Ruoergai County (e) from 2000 to 2015; Table S1: Annual rainfall events at four meteorological stations [58]; Table S2: Areas of different water retention pixel values from 2000 to 2015 based on five models; Table S3: Total water retention amount of various land types by models; Table S4: Changing areas of different pixel values in the study area from 2000 to 2015 for five models.

**Author Contributions:** Conceptualization, M.S., L.Y. and J.H.; Methodology, M.S., L.Y., Y.L. and J.H.; Software, M.S.; Data curation, X.C., L.Y. and J.H.; Writing—Review and Editing, M.S., X.C, Y.L. and J.H.; Visualization, J.H.; Supervision, J.H. All authors have read and agreed to the published version of the manuscript.

**Funding:** This study was supported by the Second Tibetan Plateau Scientific Expedition and Research of China (no. 2019QZKK0307), the Sichuan Science and Technology Program of China (no. 2022JDR0307), the National Science & Technology Fundamental Resources Investigation program of China (no. 2021FY100704), the National Natural Science Foundation of China (no. 42007057), and the Fundamental Research Funds for the Central Universities, Southwest Minzu University, China (no. 2021NYYXS14).

**Data Availability Statement:** Not applicable.

**Acknowledgments:** We thank anonymous reviewers for their original insights. We are also immensely grateful to the editor for their comments on the manuscript.

**Conflicts of Interest:** The authors declare no conflict of interest.

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
