# Peer review of "Comparison of Five Models for Estimating the Water Retention Service of a Typical Alpine Wetland Region in the Qinghai–Tibetan Plateau"

_remotesensing, doi:10.3390/rs14246306_

Round 1

Reviewer 1 Report

This study analyzed the changes in spatial and temporal patterns of water retention service and the driving factors using five different models on the Ramsar international alpine wetland region of Qinghai-Tibetan Plateau, which is rich in content and reliable in results, but the content needs further revision.

1.        This study used partial correlation analysis to explain the drivers related to WR service in the alpine wetland region. Please put the T-test results so that the reader can see the region of significant correlation.

2.        The explanation for the differences in the correlation between the five different models and the drivers is not sufficient.

3.        There is too little content in the discussion of the impact of human disturbance on runoff, and since this study already has the analysis of land use change, it is recommended to add the impact of land use change from a landscape perspective.

4.        The two green colors in Figure 6 are not obvious, please replace them. Also, please indicate which result corresponds to each legend color before “positive correlation and negative correlation” in the legend.

5.        Please pay attention to grammar, spelling, and sentence structure of manuscript so that the goals and results of the study are clear to the reader.

Author Response

We have revised the manuscript considerably. Firstly, land use distribution map from 2000 to 2015 was added to the study area, and land use transfer analysis have been added to the first part of the results. Secongly, change trends analysis in temperature, precipitation, evapotranspiration and effective energy and mass transfer were added in the second part of the results. Thirdly, multiple comparative analyses of the evaluation results of different models are added. Finally, all modified sections were re-discussed and changed.

The suggestions, questions, and responses are listed as follows:

Point 1: This study used partial correlation analysis to explain the drivers related to WR service in the alpine wetland region. Please put the T-test results so that the reader can see the region of significant correlation.

Response 1:

Thanks for the reviewer’s comments. According to the T-test formula (Eq. 13) in the manuscript, the positive and negative regions in Figure 8 and Figure 9 were the significant regions at the P<0.05 level after the T-test. We modified this part in our manuscript.

Point 2: The explanation for the differences in the correlation between the five different models and the drivers is not sufficient.

Response 2:

Thanks for the reviewer’s comments. We analyzed change trends of precipitation (P), temperature (T), and Effective energy and mass transfer (EEMT) from 1980 to 2018 and evapotranspiration (ET) from 2000 to 2018 based on a linear regression method. Then we added some discussion about the difference between models and natural drivers. In addition, land use change is a human-dominated social-economic driver that has a critically impacts on WR services. Therefore we also added to the discussion of the impact of land use change on WR services.

Point 3: There is too little content in the discussion of the impact sof human disturbance on runoff, and since this study already has the analysis of land use change, it is recommended to add the impact of land use change from a landscape perspective.

Response 3:

Thank you for your suggestion sincerely. We agreed to improve our manuscript, added more details, and incorporated this change in the revised manuscript.

Point 4: The two green colors in Figure 6 are not obvious, please replace them. Also, please indicate which result corresponds to each legend color before “positive correlation and negative correlation” in the legend.

Response 4:

Thanks for the reviewer’s advice. We have changed the colors in Figure 8, and relocated the legend from the bottom to the right region so that the correlation figures correspond to the legend in Figure 8 and Figure 9.

Point 5: Please pay attention to grammar, spelling, and sentence structure of manuscript so that the goals and results of the study are clear to the reader.

Response 5:

Thanks for the reviewer’s suggestion sincerely. We have checked all the expressions in the manuscript carefully.

Reviewer 2 Report

First, this work did not use remote sensing technology, so why was it submitted to Remote Sensing?

The contents are just exercises of using existing models without obtaining new scientific insights. 

The title indicates that it is a study about peatland. However, the simulations did not consider the characteristics of peatland.  The explanations about the underestimation and overestimation with different models did make any sense without considering field conditions. 

Author Response

We have revised the manuscript considerably. Firstly, land use distribution map from 2000 to 2015 was added to the study area, and land use transfer analysis have been added to the first part of the results. Secongly, change trends analysis in temperature, precipitation, evapotranspiration and effective energy and mass transfer were added in the second part of the results. Thirdly, multiple comparative analyses of the evaluation results of different models are added. Finally, all modified sections were re-discussed and changed.

The suggestions, questions, and responses are listed as follows:

Point 1: First, this work did not use remote sensing technology, so why was it submitted to Remote Sensing?

Response 1:

Thanks for the reviewer’s comments. Firstly, we used a 30 m high-resolution land use remote sensing data source obtained from the internal shared data of the Second Tibetan Plateau Scientific Expedition and Research of China. The Qinghai-Tibetan Plateau, named the Third Pole, with high altitude, thick cloud cover, and complex surface morphology, making it challenging to produce and obtain high-resolution land use data. The land use data were built explicitly for the Qinghai-Tibetan Plateau, ensuring the original data’s accuracy. Secondly, through spatial methods, we applied high-resolution remote sensing data to the world water tower region, a hot region of hydrology and water resources research. We need to use specific remote sensing knowledge and technology in statistical analysis, spatial mapping, and simulation of remote sensing data.

Point 2: The contents are just exercises of using existing models without obtaining new scientific insights.

Response 2:

Thanks for the reviewer’s comments. We compared the performance of five different structured WR service evaluation models, including three types of water balance-based, process-based, and surrogate biophysical indicators-based in the alpine wetland area. We found that WR service by water balance-based and process-based models all showed a spatial pattern consistent with rainfall, which indicates that models related to hydrological processes have a high degree of consistency in revealing spatial and temporal patterns of WR. In addition, we found the inapplicability of general model parameters to the study area. At the same time, we also found that the surrogate biophysical indicators-based revealed the ability of the WR service of herbaceous marsh wetlands, which is undoubtedly an essential aspect of alpine wetlands. These findings could help clarify the applicability of the water retention models in an alpine wetland region and provide a specific theoretical basis for selecting the most suitable model of water retention service at a regional or global scale.

Point 3:  The title indicates that it is a study about peatland. However, the simulations did not consider the characteristics of peatland. The explanations about the underestimation and overestimation with different models did make any sense without considering field conditions.

Response 3:

Thanks for the reviewer’s comments. We adjusted our manuscript title to “Comparison of five models for estimating water retention service of a typical alpine wetland region in the Qinghai-Tibetan Plateau”. On the one hand, alpine peatlands are essential for providing water-related services. Zoige Wetland is an integral part of the international Ramsar wetland and a critical distribution area of alpine peatland. The geographical location of the eastern margin of the Qinghai-Tibetan Plateau and the upper reaches of the Yellow River makes this region provides extremely vital water-related services, such as water retention and hydrological regulation. Therefore, an accurate assessment of water-related services and their changes will contribute to rational planning and improvement of ecosystem services, so we selected this area as the study area. On the other hand, evaluation models for WR services have been widely used worldwide, but few models are unique to the particular land type of peatland. Our results showed how different models’ performance on peatlands. According to the model’s performance, the hydrological processes involved in peatlands are discussed, and the future direction of remote sensing technology in peat wetlands is proposed.

Reviewer 3 Report

Dear authors,

Please check the comments in the attached document.

Author Response

We have revised the manuscript considerably. Firstly, land use distribution map from 2000 to 2015 was added to the study area, and land use transfer analysis have been added to the first part of the results. Secongly, change trends analysis in temperature, precipitation, evapotranspiration and effective energy and mass transfer were added in the second part of the results. Thirdly, multiple comparative analyses of the evaluation results of different models are added. Finally, all modified sections were re-discussed and changed.

The suggestions, questions, and responses are listed as follows:

Point 1: The definition of WR service is quite unclear. First, it seems that this study focuses on “water storage change” rather than absolute water storage (e.g. WAB I). This should be mentioned in the manuscript. Second, the definition of WR service differed in different models, which makes the results uncomparable. Third, the unit of WR is not consistent (e.g. mm in WAB I; m3 in WAB II; dimensionless in NBS model), the how to draw them into one figure (e.g. Figure 5). In addition, eq. 8 is not correct (from the unit, km2*mm*10-3 is not m3).

Response 1:

Thanks for the reviewer’s comments. We modified the description of five models of WR service more suitable in our manuscript. We used five widely used models with different understandings of water retention (WR) services, and the differences in the units of the results were difficult to analyze at the same level. Therefore, we paid more attention to the spatial pattern and WR change within the same year or the same temporal scales instead of quantity, which is mentioned in the introduction. For example, in the results, except for the NBS model, the other four models all showed similar spatial patterns of WR services, which could also reflect the similarities and differences between the models. The unit is dimensionless except for the NBS model, and the other four models are measured in mm or m3 among the five models. We used the resolution of the raster pixel (30 m) to unify the results of the four models. There is no unit in the figures, and we explain them in the note below figures. In addition, we have corrected formula 8 and Figure 6 in the manuscript.

Point 2: The uncertainty of the results is not shown. Uncertainty is one important component of a model, which should be shown together with the mean value. The authors gave a discussion on the uncertainty of the model. But still, the estimation of the individual model needs uncertainty (e.g. value range).

Response 2:

Thanks for the reviewer’s comments. We accepted this advice and revised our manuscript. According to the literature, we adopt the following methods: Firstly, we randomly created 100 sample points within the study area and extracted the WR attribute values by models of the corresponding spatial locations in 2000, 2010, and 2015 based on the spatial analysis tool. Secondly, we calculated the mean values of the same model as group data for a model because the ordering relationship of WR service remains the same in the same year. Thirdly, extreme values in a group were eliminated to make the mean and standard deviation within a reasonable range. Each model kept 83 points. Fourthly, we draw a mean-standard deviation bar graph to show the uncertainty of the model. At the same time, to show the statistical differences of WR services evaluated by different models, one-way ANOVA and multiple comparisons were performed on the evaluation results, and we distinguished the NBS model from the other four models.

References:

[1]    Ye,Y.C.; Zhou, G.H; Yin, X.J. Changes in distribution and productivity of steppe vegetation in Inner Mongolia during 1961 to 2010: Analysis based on MaxEnt model and synthetic model.Acta Ecologica Sinica, 2016, 36, 4718-4728, doi: 10.5846/stxb201412302599.

[2]    Li, H.Y. Research on Water Conservation Function of Xiaoxing’an Mountains Based on GIS. Territory & Nature Resources Study. 2021, 2, 50-53, dio: 10.16202/j.cnki.tnrs.2021.02.015.

[3]    Li, Y.Y.; Ma, X.S.; Qi, G.H.; Wu, Y.L. Studies on Water Retention Function of Anhui Procince Based on InVEST Model of Parameter Localization. Resources and Environment in the Yangtze Basin. 2022, 31, 313-325,doi: 10. 11870 /cjlyzyyhj202202006.

Point 3: Some input parameters are quite unclear. (1) It is unclear how the authors derive the input parameters. For instance, the soil hydraulic conductivity (cm/d), calculated by Neuro Theta software?? This information is quite unclear. Given the fact that the soil hydraulic conductivity is critical in runoff estimation. The authors should make it clear. (2) For the INVEST model and NBS model, please do a parameter sensitivity analysis. The authors should report that the results are sensitive to which parameters the most.

Response 3:

Thanks for the reviewer’s comments. We added detailed information to describe selected models and input parameters. In discussing the uncertainties and driving factors of InVEST and NBS models, we cited the analysis results in the literature[1-3], where researchers have reported the effects of parameters in both models on WR service evaluation results. The references were as fellows.

References:

  1. Wang, Y.F.; Ye, A.Z.; Peng, D.Z.; Miao, C.Y.; Di, Z.H.; Gong, W. Spatiotemporal variations in water conservation function of the Tibetan Plateau under climate change based on InVEST model. J. Hydrol. Reg. Stud. 2022, 41, 101064, doi:10.1016/j.ejrh.2022.101064.
  2. Zhang, L.W.; Lü, Y.H.; Fu, B.J.; Dong, Z.B.; Zeng, Y.; Wu, B.F. Mapping ecosystem services for China's ecoregions with a biophysical surrogate approach. Landsc. Urban. Plan. 2017, 161, 22-31, doi:10.1016/j.landurbplan.2016.12.015.
  3. Zheng, H.B.; Zhang, L.W.; Wang, P.T.; Li, Y.J. The NPP-Based Composite Indicator for Assessing the Variations of Water Provision Services at the National Scale. Water 2019, 11, 8, doi:10.3390/w11081628.

Point 4: The Discussion section is quite weak. (1) The authors showed the social factors in the Results section but not in the discussion section. (2) The drivers of the WR service are not well explained and the general trend of precipitation, ET, and runoff of this study area is missing. (3) It is unclear the dynamics of land management and how it may change the WR service. (4) Lines 445-451, the information here is unclear, “EEMT is negatively in most wetland parts by the PRS, InVEST, and NBS models while positively in the WAB I and WAB â…¡ models”. Detained information should be given and the process behind should be discussed.

Response 4: 

Thanks for the reviewer’s careful comments. In the Discussion section, we have added more content and incorporated this change in the revised text. Details were as follows:

  • the authors showed the social factors in the Results section but not in the discussion section.

Land use change is a critical human-dominant social-economic driver that has a critical impact on WR services, therefore, we also added to the discussion of the impact of land use change on WR services.

(2) The drivers of the WR service are not well explained and the general trend of precipitation, ET, and runoff of this study area is missing.

We analyzed change trends of precipitation (P), temperature (T), and effective energy and mass transfer (EEMT) from 1980 to 2018 and evapotranspiration (ET) from 2000 to 2018 based on a linear regression method further. Then we added some discussion about the difference in spatial aspects between models and natural drivers.

(3) It is unclear the dynamics of land management and how it may change the WR service.

We added the analysis of land use change and land transfer in the study area from 2000 to 2015. At the same time, we discussed the changes in land use types caused by major human activities in the study area and their effects on WR services.

(4) Lines 445-451, the information here is unclear, “EEMT is negatively in most wetland parts by the PRS, InVEST, and NBS models while positively in the WAB I and WAB â…¡ models”. Detained information should be given and the process behind should be discussed.

Thanks for the reviewer’s careful comments. In this part, we didn’t accurately describe the relationship between EEMT and WR service, and the differences between models. So, we revised this part by using a more proper discussion in the main text.

Round 2

Reviewer 1 Report

The author has revised it as requested and suggested it for acceptance

Author Response

We have promoted our manuscript through the editor's and reviewers' comments. Firstly, we corrected the main reasons for the lowest evaluation of wetland water retention services by the InVEST model due to some misleading in evaluating baseflow in the first part of discussion and abstract. Although we considered this reason, in the version (InVEST 3.9.0), we applied, and in a newer version, the effect of the baseflow process on water retention has been taken into account. Secondly, we added the explanation of difficulties and uncertainties arising from acquiring long temporal and high spatial resolution remote sensing data for evaluating WR services at the alpine region in the first part of discussion. Thirdly, we polished the English language, grammar, punctuation, spelling, and overall style of the text with the help of the highly qualified native English speaking editors at AJE and incorporated all the changes in the revised manuscript. The manuscript's spelling, grammar, and sentence errors have been carefully checked and revised. Finally, we have carefully checked and updated the references according to the journal's standards.

Response to Reviewer Comments

The suggestions, questions, and responses are listed as follows:

Point 1:

The author has revised it as requested and suggested it for acceptance

Response 1:

Thanks for the recognition on our revised manuscript.

Reviewer 2 Report

Dear Authors

Thank you for your efforts to improve the manuscript. However, a few comments are given below

1. End-use of remote sensing data does not justify the submission to Remote Sensing unless there is an innovation.

2. Your results showed different model performances on peatland. Could you explain why in consideration of peatland characteristics?

Author Response

We have promoted our manuscript through the editor's and reviewers' comments. Firstly, we corrected the main reasons for the lowest evaluation of wetland water retention services by the InVEST model due to some misleading in evaluating baseflow in the first part of discussion and abstract. Although we considered this reason, in the version (InVEST 3.9.0), we applied, and in a newer version, the effect of the baseflow process on water retention has been taken into account. Secondly, we added the explanation of difficulties and uncertainties arising from acquiring long temporal and high spatial resolution remote sensing data for evaluating WR services at the alpine region in the first part of discussion. Thirdly, we polished the English language, grammar, punctuation, spelling, and overall style of the text with the help of the highly qualified native English speaking editors at AJE and incorporated all the changes in the revised manuscript. The manuscript's spelling, grammar, and sentence errors have been carefully checked and revised. Finally, we have carefully checked and updated the references according to the journal's standards.

Response to Reviewer Comments

The suggestions, questions, and responses are listed as follows:

Point 1: End-use of remote sensing data does not justify the submission to Remote Sensing unless there is an innovation.

Response 1:

Thanks for the reviewer’s comments. In this study, we applied high-resolution land use data of the Qinghai-Tibetan Plateau to simulate the performance of different models in evaluating alpine wetland water retention services. This work has a vital reference for selecting suitable models to improve the evaluation of water retention services at a small regional scale. Land use/land cover(LULC) products at different spatial scales (local, national, and global) are essential for remote sensing ecological assessment research. However, current land use products at the global or national scale with high resolution are still unable to subdivide marshes, herbaceous wetlands, or peatlands regionally. For example, the GlobeLand30 dataset (http://www.globallandcover.com/) was divided into a wetland with plants and water bodies and a 30 m annual land cover dataset (https://essd.copernicus.org/articles/13/3907/2021/) only classified wetlands as the first category. Although the study area has more prominent high-altitude landscape heterogeneity, more complex landforms, and more prosperous vegetation communities and ecosystem types, existing products are unable to fully distinguish and describe the LULC variations at the level of ecosystem types in the region [1]. Some studies have considered more accurate land use classification in the study area. They classified the other wetland types as marsh, marshy meadow, and peatland, except for lakes and rivers [1-6]. However, the current high-quality spatial data still cannot meet the needs of a wide range of applications because when the spatial resolution is higher, the time resolution cannot be continuous.The land use data we used in this study divided wetlands into four secondary categories: herbaceous wetland, lake, river, and reservoir or pond. The herbaceous wetland is one of the critical land use types in retaining water sources in this study area. At the same time, the temporal scale (2000-2015) of the land use data we used was sufficient to meet the research needs.

References:

  1. Feng, S.Y.; Li, W.L.; Xu, J.; Liang, T.G.; Ma, X.L.; Wang, W.Y.; Yu, H.Y. Land Use/Land Cover Mapping Based on GEE for the Monitoring of Changes in Ecosystem Types in the Upper Yellow River Basin over the Tibetan Plateau. Remote. Sens. 2022, 14, 5361. https://doi.org/10.3390/ rs14215361.
  2. Dong, L.Q.; Yang, W.; Zhang, K.; Zhen, S.; Cheng, X.P.; Wu, L.H. Study of marsh wetland landscape pattern evolution on the Zoigê Plateau due to natural/human dual-effects. PeerJ 2020, 8, e9904, doi:10.7717/peerj.9904.
  3. Hu, X.D.; Zhang, P.L.; Zhang, Q.; Wang, J.Q. Improving wetland cover classification using artificial neural networks with ensemble techniques. Gisci Remote Sens. 2021, 58, 603-623, doi:10.1080/15481603.2021.1932126.
  4. Li, Z.W.; Gao, P.; Hu, X.Y.; Yi, Y.J.; Pan, B.Z.; You, Y.C. Coupled impact of decadal precipitation and evapotranspiration on peatland degradation in the Zoige basin, China. Phys Geogr 2020, 41, 145-168, doi:10.1080/02723646.2019.1620579.
  5. Shen, G.; Yang, X.C.; Jin, Y.X.; Luo, S.; Xu, B.; Zhou, Q.B. Land Use Changes in the Zoige Plateau Based on the Object-Oriented Method and Their Effects on Landscape Patterns. Remote Sens. 2020, 12, 14, doi:10.3390/rs12010014.
  6. Gao, C.; Huang, C.; Wang, J.B.; Li, Z. Modelling Dynamic Hydrological Connectivity in the Zoigê Area (China) Based on Multi-Temporal Surface Water Observation. Remote Sens. 2022, 14, 145, doi:10.3390/rs14010145.

Point 2: Your results showed different model performances on peatland. Could you explain why in consideration of peatland characteristics?

Response 2:

Thanks for the reviewer’s comments. In our description of the five models in this study, except for the WAB I and NBS models, which only need to input spatial data, the other three must input parameters matching land use types. For example, the biophysical table in the InVEST model, the runoff ratio in the PRS model, and the surface runoff coefficient in the WAB II model. For such parameters, we have used the land use secondary classification as a reference, taking into account the properties of peatland or herbaceous wetlands and choosing localized parameters where possible. In addition, the NPP data input in the NBS model also well revealed the spatial characteristics of the peatland wetland in this alpine wetland region in the third part of the results. At the same time, we also discuss the uncertainty of the vegetation-covered marshland of the plateau region in remote sensing information recognition. The Zoige Plateau is a typical alpine wetland water retention area. Our study revealed the performance of the general model in this area, which provides some reference for promoting the development of water retention mechanism and evaluation model of peatland in the future.

Reviewer 3 Report

Thanks!

Author Response

We have promoted our manuscript through the editor's and reviewers' comments. Firstly, we corrected the main reasons for the lowest evaluation of wetland water retention services by the InVEST model due to some misleading in evaluating baseflow in the first part of discussion and abstract. Although we considered this reason, in the version (InVEST 3.9.0), we applied, and in a newer version, the effect of the baseflow process on water retention has been taken into account. Secondly, we added the explanation of difficulties and uncertainties arising from acquiring long temporal and high spatial resolution remote sensing data for evaluating WR services at the alpine region in the first part of discussion. Thirdly, we polished the English language, grammar, punctuation, spelling, and overall style of the text with the help of the highly qualified native English speaking editors at AJE and incorporated all the changes in the revised manuscript. The manuscript's spelling, grammar, and sentence errors have been carefully checked and revised. Finally, we have carefully checked and updated the references according to the journal's standards.

Response to Reviewer Comments

The suggestions, questions, and responses are listed as follows:

Point 1: Thanks!

Response 1:

Thanks for the recognition on our revised manuscript.

Round 3

Reviewer 2 Report

The manuscript was Improved.